# Association between Heavy Metals and Rare Earth Elements with Acute Ischemic Stroke: A Case-Control Study Conducted in the Canary Islands (Spain)

**DOI:** 10.3390/toxics8030066

**Published:** 2020-09-02

**Authors:** Florián Medina-Estévez, Manuel Zumbado, Octavio P. Luzardo, Ángel Rodríguez-Hernández, Luis D. Boada, Fernando Fernández-Fuertes, María Elvira Santandreu-Jimenez, Luis Alberto Henríquez-Hernández

**Affiliations:** 1Rehabilitation Service, Complejo Hospitalario Insular-Materno Infantil (CHUIMI), Avenida Marítima del Sur, 35016 Las Palmas de Gran Canaria, Spain; florianmed@gmail.com (F.M.-E.); fferfue@gobiernodecanarias.org (F.F.-F.); maviras@yahoo.es (M.E.S.-J.); 2Toxicology Unit, Research Institute of Biomedical and Health Sciences (IUIBS), Department of Clinical Sciences, Universidad de Las Palmas de Gran Canaria (ULPGC), Paseo Blas Cabrera Felipe s/n, 35016 Las Palmas de Gran Canaria, Spain; manuel.zumbado@ulpgc.es (M.Z.); octavio.perez@ulpgc.es (O.P.L.); anrodrivet@gmail.com (Á.R.-H.); luis.boada@ulpgc.es (L.D.B.)

**Keywords:** stroke, cerebrovascular accident, heavy metal, rare earth element, case-control study

## Abstract

The role of inorganic elements as risk factors for stroke has been suggested. We designed a case-control study to explore the role of 45 inorganic elements as factors associated with stroke in 92 patients and 83 controls. Nineteen elements were detected in >80% of patients and 21 were detected in >80% of controls. Blood level of lead was significantly higher among patients (11.2 vs. 9.03 ng/mL) while gold and cerium were significantly higher among controls (0.013 vs. 0.007 ng/mL; and 18.0 vs. 15.0 ng/mL). Lead was associated with stroke in univariate and multivariate analysis (OR = 1.65 (95% CI, 1.09–2.50) and OR = 1.91 (95% CI, 1.20–3.04), respectively). Gold and cerium showed an inverse association with stroke in multivariate analysis (OR = 0.81 (95% CI, 0.69–0.95) and OR = 0.50 (95% CI, 0.31–0.78)). Future studies are needed to elucidate the potential sources of exposure and disclose the mechanisms of action.

## 1. Introduction

Ischemic stroke is a sudden disorder of cerebral blood flow that temporarily or permanently alters the function of a certain region of the brain. According to the World Stroke Organization (WSO), age-adjusted rate for ischemic stroke per 100,000 populations was 142.34 in 2016. Over 9.5 million new cases were diagnosed during that year [1]. Prevalence of stroke in US population younger than 60 years old is around 2%, but this proportion rises to 6% and 15% in people older than 60 and 80 years old, respectively [2]. In Spain, it has been estimated a prevalence of stroke of 6.4% in subjects older than 70 years [3]. 

Age-adjusted incidence ranges from 50 to 250 per 100,000 populations in France and Portugal, respectively [4]. In our country, for a similar period, incidence ranged from 99 to 206 per 100,000 populations, depending on the region in which the study was conducted [4]. The Canary Islands are the second region of Spain with the lowest adjusted rate of stroke (25.33 per 100,000 men and 19.66 per 100,000 women) [3].

Stroke is the second leading cause of death worldwide, and the third leading cause of disability [5]. In Spain, mortality has decreased considerably during the past decades, standing at around 50 per 100,000 inhabits for both sexes [6]. The Canary Islands are the second autonomous community in Spain with the lowest adjusted rate of cerebrovascular deaths (25.3 per 100,000 males and 19.7 per 100,000 women) [3]. 

The main risk factors for stroke are mostly modifiable factors such as hypertension, dyslipidemia, diabetes, smoking, low physical activity levels, unhealthy diet and abdominal obesity [2]. Age, gender, race/ethnicity or genetics also have an important role for the disease [7]. New risk factors for stroke have been proposed during the last decades. Some of them have emerged as protective factors (i.e., antiplatelet therapy) while others seem to increase the risk of stroke (i.e., sleep apnea or lipoprotein levels) [2]. However, their contribution to stroke risk is less well defined and understood. This is the case of environmental pollutants. In addition to gaseous and particulate air pollutants [8], persistent organic pollutants (POPs) seem to play an important role in development of stroke [9]. Although the association had been seen for years, it has recently been observed that elevated serum POPs levels were associated with an increased risk of stroke, specifically for organochlorine pesticides (p,p’-DDE) and polychlorinated biphenyls (PCB-118, -156 and -138) [10], possibly due to an association with hypertension [11,12] and obesity [13]. 

Among environmental pollutants, toxic heavy metals and metalloids are among the most dangerous because they are also not biodegradable and tend to accumulate in environmental compartments [14]. According to their high degree of toxicity, arsenic, cadmium, lead and mercury are usually highlighted among others [14]. However, the Agency for Toxic Substances and Disease Registry (ATSDR) publishes a list of priority chemicals that are determined to represent the most significant potential threat to human health because of their known or suspected toxicity, together with the potential human exposure. Additionally, there are a number of elements, the rare earth elements (REE) and other minor elements (ME), which are increasingly coveted due to the large number of technological applications for which they are already indispensable [15]. This set of elements is of growing concern because its enormous range of applications makes them mobilized from the few sites where they are abundant to be distributed all over the planet [16], especially once the useful life of the devices containing them ends. Although some of these elements are relatively abundant in the Earth’s crust (i.e., cerium is as abundant as copper), REEs have been included among the new and emerging occupational and environmental health risks by several international organizations [17]. Different studies have shown that some of these elements have an adverse effect on people’s health [18,19,20], although the mechanisms of action are not clear [17,21]. 

Of all these inorganic elements, arsenic and lead have shown a relationship with stroke [22,23,24,25]. A significant dose-response relationship was observed between arsenic concentration in well water and prevalence of cerebrovascular disease [25] and a positive trend was reported between blood lead and stroke in a series of 88,000 workers from USA, Finland and UK [24]. However, the mechanisms of action are not clear and, for example, a potential role for arsenic methylation in the pathogenesis of stroke has been suggested [22].

It has to be highlighted that some inorganic elements are neurological disruptors with the capacity to cross the blood brain barrier [18]. While small concentrations of some elements are needed for life, most are considered non-essential and some are very toxic even at very low concentrations. Some elements follow a hormetic dose–response curve and may cause, at a very low dose, the opposite effect to a high dose [26]. Thus, the presence of these elements into neurons, even at low concentrations, seems to be able to modify brain homeostasis. This is the case of gadolinium and tantalum whose tissue concentrations were higher among patients with brain cancer compared to a control group [18]. The aim of the present study was to evaluate the contribution of 45 inorganic elements—including trace elements, elements included in the priority list of substances of the Agency for Toxic Substances and Disease Registry (ATSDR), and REE and other elements used in electronic devices—as factors associated with stroke.

## 2. Patients and Methods

### 2.1. Study Design and Participants

We designed a case-control study aimed to disclose the role of inorganic elements in the stroke. The study was approved by the Research Ethics Committee of the CHUIMI on 23 February 2017 (ID number: CEIm-CHUIMI-2017/907). For that, a total of 92 patients diagnosed with ischemic stroke and admitted into the Complejo Hospitalario Insular-Materno Infantil CHUIMI for rehabilitation were included in the study. The control group consisted of patients admitted to the rehabilitation service for other causes. The final number of control subjects was 83. The recruitment was made between April 2017 and April 2018. The inclusion criteria for cases were: (i) having been diagnosed of stroke in the 12 months prior to being referred to the Rehabilitation Service, (ii) ability to agree to participate in the study (signed informed consent) and (iii) being over 18 years old. The inclusion criteria for controls were: (i) admission diagnosis unrelated with stroke (i.e., traumatic diseases), (ii) never having been diagnosed with stroke, (iii) ability to agree to participate in the study (signed informed consent) and (iv) being over 18 years. Cases and controls were matched in terms of age and gender making a selection of cases in relation to the demographic characteristics of the cases.

Patients—cases and controls—were contacted and asked to participate in the study. All patients signed the informed consent before entering the study. The study was approved by the Research Ethics Committee of the CHUIMI (study number CEIm-CHUIMI-2017/907).

Barthel index, an instrument widely used to evaluate independency and measures the capacity of the person for the execution of ten basic activities in daily life [27], was recorded three months after the admission in the rehabilitation service. The demographic and clinical data of the patients included in the study were collected from the corresponding medical records and are shown in Table 1.

Blood samples were obtained from all of the participants. Samples of blood were collected in 4 mL heparinized tubes (BD Vacutainer, LH 68 I.U. Lithium Heparin, BD-Plymouth, PL6 7BP, UK) and maintained at 4 °C. An aliquot of blood was stored at −80 °C until the chemical analysis, performed in the Toxicology Unit of the ULPGC.

### 2.2. Selection of Elements and Sample Preparation

Blood concentration levels of 45 inorganic elements were analyzed. We determined trace elements, heavy metals, rare earth elements (REEs) and other elements used in electronic devices, as we have previously reported [15,28].

Then, 100 mg of whole blood was weighed into quartz digestion tubes and then digested into 1 mL of acid solution (65% HNO_3_) using a Milestone Ethos Up equipment (Milestone, Bologna, Italy). The digestion conditions were programmed as follows (power (W)–temperature (°C)–time (min): step 1: 1800–100–5; step 2: 1800–150–5; step 3: 1800–200–8; and step 4: 1800–200–7. After cooling, the digested samples were transferred and diluted. An aliquot of each sample was taken and the internal standard (ISTD) was added for the analysis.

The ISTD solution included scandium, germanium, rhodium and iridium (20 mg/mL each). Elements of standard purity (5% HNO_3_, 100 mg/L) were purchased from CPA Chem (Stara Zagora, Bulgaria). Two standard curves (range = 0.005–20 ng/mL) were made: (a) one used a commercial multi-element mixture (CPA Chem Catalog number E5B8·K1.5N.L1, 21 elements) containing all the trace elements and the main heavy metals and (b) the other multi-element mixture included individual elements (CPA Chem) that contained the REEs and other elements used in electronic devices [18,19].

### 2.3. Analytical Procedure

An Agilent 7900 ICP-MS (Agilent Technologies, Tokyo, Japan) was used for quantification. Instrument configuration and optimization were previously reported [18]. The elements were quantified in the MassHunter v.4.2. ICP-MS Data Analysis software (Agilent Technologies).

The analytical method was previously validated [29]. The recoveries rates were 89–128% for REEs and other elements used in electronic devices, and 87—118% for ATSDR’s toxic heavy elements and trace elements (regression coefficients > 0.998 for all elements). The limit of quantification (LOQ) was calculated by quantifying 6 replicates of blanks, consisting in 0.130 μL of alkaline solution, as the concentration of the element that produced a signal three times higher than that of the averaged blanks (Appendix A). The accuracy and precision were assessed by substituting the sample with a fortified alkaline solution (0.05, 0.5 and 5 ng/mL). The calculated relative standard deviations (RSD) were lower than 8%, except for copper, nickel, selenium, iron, barium, zinc and samarium. However, at the lowest level of fortification, the RSD was higher than 15–16%.

### 2.4. Statistical Analysis

Descriptive analyses were conducted for all of the variables. Medians, ranges and the 5th–95th percentiles of the distribution were calculated for continuous variables. Proportions were calculated for categorical variables. Values below the LOQ were assigned a random value between 0 and the LOQ [18,30]. For this, a specific computational function was used (Microsoft Excel (2010), RANDBETWEEN function).

The normality of the data was assessed using the Kolmogorov–Smirnov test. Since most of the data (concentrations of elements) did not follow a normal distribution, comparisons between groups were performed using non-parametric tests (Kruskal–Wallis and Mann–Whitney U-test). Differences in the categorical variables were tested with the Chi-square test. The correlation of inorganic elements with continuous variables (age and Barthel index) was analyzed with Pearson’s correlation test. Bivariate correlations among elements were done with Spearman’s rho test. Univariate and multivariate analyses were done with logistic regression test. For multivariate logistic regression analysis, smoking, arterial hypertension, dyslipidemia and coronary cardiopathy were included as covariates. These variables were specifically included because they showed to be a significant risk factor for stroke in the present series (Figure 1). Values of elements were log transformed before the inclusion in logistic regression analyses. We used PASW Statistics v 19.0 (SPSS Inc., Chicago, IL, USA) to manage the database and to perform the statistical analyses. Probability levels of <0.05 (two-tailed) were considered statistically significant.

## 3. Results and Discussion

A total of 45 inorganic elements were measured. To better understand the main results, elements were separated into two different categories: (i) trace elements and inorganic elements included in the ATSDR’s priority pollutant list, which includes heavy metal and other well-known toxic elements [28]; and (ii) RREs and other elements employed in the manufacture of electronic devices [15].

### 3.1. Clinical Characteristics of Cases and Controls

A total of 92 patients, admitted into the Rehabilitation Service after suffering a stroke, and 83 control patients were included in the study. No significant differences were detected in age and gender distribution (Table 1). Although it has been published that three quarters of strokes occur in patients over 65 [6], we observed that this proportion was 50.0% in our series (Data not shown).

The distribution of the main clinical factors associated to the stroke [7] were significantly different among cases and controls, with the exception of diabetes. Smoking, arterial hypertension, dyslipidemia and coronary cardiopathy appeared associated with stroke (Figure 1). We observed that hypertension was the factor that showed the highest significance (odds ratio (OR) = 3.86 (confidence interval (CI) 95%, 2.06–7.24), *p*-value ˂ 0.0001; univariate analysis), as previously established [7]. Regarding diabetes, it has been published a 2-fold increased risk in stroke for diabetic patients, and stroke accounts for approximately 20% of deaths in diabetics [7]. However, we did not observe this trend in our series. Canary Islands have one of the highest ratios of diabetes in Spain [31]. In that sense, while 27.2% of cases were diabetics, this proportion was higher (34.9%) among controls (Table 1). This pattern of distribution makes difficult to observe the role of diabetes in our study population. Cigarette smoking remains a major factor for stroke [32]. We absolutely agree with that observation and an OR = 2.51 (Figure 1) was observed in the present series.

Mean value of Barthel index was 67.8 and 93.2 among cases and controls, respectively (*p* ˂ 0.0001, Mann–Whitney U-test), and 41.3% of patients who suffered a stroke showed a severe dependence (Table 1). This profile is similar to other studies focused in the evaluation of disability after a stroke [33].

### 3.2. ATSDR’s Priority Elements in Stroke

A total of 19 inorganic elements including in the ATSDR’s priority pollutant list [28] were analyzed in the whole blood of cases and controls. Serum is the matrix of choice for the determination of inorganic elements, mainly for trace elements whose reference values are those of the serum. However, in an effort to prioritize the toxic elements whose presence is mostly found in blood cells (i.e., lead or mercury), whole blood was the matrix of study [18]. We observed a high frequency of detection in both groups, where most of the elements were present in more than 75% of the series (Table 2). The less frequently detected elements were beryllium in the control group (8.4%), and palladium among stroke patients (9.8%).

In general, we do not observe any influence of age on the distribution of these elements, with the exception of antimony and thorium that showed a significant positive correlation in cases and controls (Pearson’s r = 0.26, *p*-value = 0.012 and Pearson’s r = 0.24, *p*-value = 0.026, respectively; Appendix A). Among patients in the control group, women presented slightly higher levels of silver, cadmium, cobalt, copper, manganese, strontium and uranium; however, among cases, this list was reduced to beryllium, cobalt and copper (Appendix A). Since age did not seem to be a determining factor for the accumulation of these substances, these differences could be attributable to other environmental factors such as diet or unhealthy habits [34,35,36,37]. However, although it is known that the smoking habit is a source of exposure of inorganic elements, we do not observe this trend in our series (Appendix A), possibly due to the low proportion of smokers (25.7% of the series was smoker) and the lack of determining covariates such as the intensity of the habit or its duration [38].

Among the 19 inorganic elements included in the ATSDR’s priority pollutant list, 2 of them had a higher blood concentration among cases (beryllium and lead); and 2 showed higher blood concentration among the controls (barium and uranium). Of these 4 elements, 3 (barium, lead and uranium) were detected in 100% of the subjects. Given the low frequency of detection, the result referring to beryllium should be taken with caution (Table 2). In univariate analysis, barium and lead were significantly associated with stroke (OR = 0.34, 95% CI 0.19–0.60, *p*-value < 0.001; and OR = 1.65, 95% CI 1.09–2.50, *p*-value = 0.019, respectively; Table 3). No significant results were observed for uranium (Data not shown). In multivariate analysis, barium and lead kept their significant tendency (Table 3).

Neither of these two elements (barium or lead) were significantly influenced by any of the known risk factors for stroke. That is, neither in the case group nor in the group of controls, was barium or lead differentially affected by diabetes, dyslipidemia, smoking or the presence of arterial hypertension (Appendix A). From this it follows that both elements can be considered independent factors associated with stroke, something plausible given the capacity of these substances to cross the blood–brain barrier and, therefore, exert an effect on brain tissue [18]. The role that inorganic elements have in relation to stroke has been studied, above all, in relation to heavy metals and major metalloids, with disparate and, sometimes, contradictory results [39]. It has to be taken into account that some toxic elements have been associated to well-known risk factors for stroke, establishing an indirect association with the disease. This is the case of arsenic, lead and specific RREs found in indoor air pollution affecting the risk of suffering hypertension [40]; or the case of arsenic, lead, cadmium and copper, whose exposure is associated with an increased risk of cardiovascular disease [23,41].

Arsenic has been previously associated with stroke [42]. In the present study, no significant differences were observed regarding to the blood concentration of arsenic among cases and controls (Table 2), possible due to the limit size of our series. Wen et al. (2019) reported median values of arsenic—among 1277 cases and 1277 controls—of 1.48 and 1.18 ng/mL, respectively [42], being significant that subtle difference of concentration. In our series, median values of arsenic were 1.61 and 1.69 ng/mL among 92 cases and 83 controls, respectively (*p*-value = 0.546; Table 2).

Lead was associated with stroke in univariate and multivariate analysis (Table 3). In that sense, lead-exposed workers showed higher mortality rate by stroke—and other diseases, a result that supports those obtained in the present study [24]. The association of ischemic stroke and lead has been shown in other studies [41,43]. However, other studies did not report any association between lead—or arsenic—and stroke [39]. Reference values (RV95s) for arsenic and lead are 2.0 and 33 ng/mL, respectively, for adult population [44]. In the present series, 35.9% (n = 33) and 39.8% (n = 33) of cases and controls, respectively, showed values of arsenic higher than RV95s (Chi-square test, *p* = 0.641; data not shown). Nobody was above RV95s for lead. This profile of distribution of elements is similar to the general population of Spain [45].

Previous publications have observed that the levels of certain inorganic elements are higher in the control group than among stroke patients [39]. This is the case of barium in the present study (Table 2). Barium is a compound frequently used in medical tests as a contrast, which makes it necessary to know details of the clinical history that were not considered in the present study. To our knowledge, this is the first time that any type of association between barium and stroke has been observed. However, this is a modest result that would require further investigation in larger series to elucidate the mechanism of action behind this association.

### 3.3. REEs and Other Inorganic Elements in Stroke

A total of 26 rare earth elements (RREs) and other elements used in the manufacturing of high tech devices [15] were analyzed in the whole blood of cases and controls. Cerium, iron and gallium were detected in 100% of subjects (Table 4). Lutetium, tantalum, terbium and thulium were detected in less than 15% of cases and/or controls. We did not observe any influence of age in relation to the blood concentration of these elements among controls. However, we observed a positive correlation of some of these elements with age among cases (Appendix A): dysprosium (Pearson’s *r* = 0.26, *p*-value = 0.013), erbium (Pearson’s *r* = 0.30, *p*-value = 0.003), europium (Pearson’s *r* = 0.25, *p*-value = 0.014), holmium (Pearson’s *r* = 0.29, *p*-value = 0.005), neodymium (Pearson’s *r* = 0.28, *p*-value = 0.008), praseodymium (Pearson’s *r* = 0.24, *p*-value = 0.020), thulium (Pearson’s *r* = 0.23, *p*-value = 0.024), yttrium (Pearson’s *r* = 0.23, *p*-value = 0.027) and ytterbium (Pearson’s *r* = 0.26, *p*-value = 0.014). Blood concentration of iron was significantly lower among women, in cases and controls (270.4 vs. 292.4 ng/mL, *p*-value = 0.002; 264.3 vs. 301.1 ng/mL, *p*-value = 0.004; respectively). We did not detect significant differences between RREs and clinical variables (Appendix A).

Of the 26 RREs, 12 showed a statistically different blood concentration between cases and controls. However, trying to guarantee a minimum statistical power, only elements with detection frequencies higher than 80% were considered. Thus, blood concentration of bismuth, cerium, gallium and osmium were higher among controls (Table 4). In univariate analysis, cerium and gallium showed an association with stroke (Table 3). No significant results were observed for bismuth and osmium (data not shown). In multivariate analysis, cerium and gallium kept their significant tendency (Table 3), which suggests that, apart from the ability to cross the blood–brain barrier [18], these elements could play a protective effect on stroke. The effect that gallium may have on stroke is difficult to assess since it is usually used as a contrast in various medical tests. It is necessary to know details of the clinical history to be able to discriminate the true effect of the association observed in the present study.

The role of these minority elements seems to be more important than initially thought. Thus, it has been recently published that gold nanoclusters penetrate the blood−brain barrier and have neuroprotective effects, suggesting the possibility of utilizing this nanoparticles to regulate microglial polarization and improve neuronal regeneration in central nervous system [46]. In the present study, blood concentration of gold was significantly higher among controls (Table 4) and the association with stroke was also found in multivariate analysis (Table 3). Although the frequency of detection of gold did not meet the quality standards imposed to guarantee a minimum statistical power, the present result agrees with others which suggest that gold is an interesting factor to consider for the treatment of stroke [46]. The neuroprotective role of cerium has been previously reported [47,48]. Cerium oxide nanoparticles, known as nanoceria, show a promising potential in diverse disorders such as stroke. The mechanism behind this effect is closely related to the antioxidant capacity of these particles [49]. Thus, the neuroprotective effects of nanoceria are due to a modest reduction in reactive oxygen species and to a reduction of the levels of ischemia-induced 3-nitrotyrosine, a modification to tyrosine residues in proteins induced by the peroxynitrite radical [47]. Optimal doses of nanoceria reduce infarct volumes and the rate of ischemic cell death [48,49] and may be useful as a therapeutic intervention to reduce oxidative and nitrosative damage after a stroke [47]. The findings observed in this regard in our series may contribute to improve the knowledge about the role of gold and cerium in relation to stroke.

### 3.4. Strengths and Limitations of the Study

The present study is a case control study aimed to evaluate the role of inorganic elements in stroke. One of the main limitations for this type of studies is the design of the groups. In that sense, we tried to minimize the impact of non-modifiable risk factors for stroke. Thus, gender and age were comparable among cases and controls. However, modifiable risk factors were different between groups, which suppose a bias that must be taken into account when interpreting the results. Ideally, the control group should exclude patients with hypertension, dyslipidemia, smokers and other obvious risk factors for stroke. Despite this, the fact that the main results were not influenced by these types of factors lends credibility to them. Sample size is a clear limitation in this type of studies. Our series included 92 cases and 83 controls, a modest number that can limit the statistical confidence. However, while it is true that similar studies have been done with a greater number of patients, other studies included smaller patient groups [39]. In any case, we tried to increase the statistical confidence by performing multivariate analysis—taken into account cofounding variables—, elements were included in the analyses after log transformation and we considered elements that showed high detection frequencies (>80%). However, we are aware that variables such as diet, details about smoking habit (intensity, duration type of tobacco and even label, which could be a significant source of inorganic elements [36]), other toxic habits like alcohol or illicit drugs intake, clinical endpoints associated with stroke (medical tests and other clinical variables like blood pressure), pharmacological treatments (antihypertensive drugs among controls) and other variables related with lifestyle (sedentary lifestyle) were avoided and could be of relevance. Similarly, we do not know the combined effect that these elements may have on human health, especially considering that exposure to many of these elements correlates with exposure to others [45]. We observed a significant amount of correlations between the elements, most of them positive (Appendix A). Moreover, the patterns of correlation appeared to be different in cases and controls (Appendix A, see correlation maps), suggesting the existence of different exposure profiles [37,50]. This finding is similar to previous published studies [45,50] and encourage exploring the combined action of contaminants. Finally, due to the characteristics of the study design, the mechanism of action behind our results can only be hypothesized. Therefore, the present study should be considered as a hypothesis generator.

According to the analyses carried out, the series seems robust both in its conformation and in its distribution, which gives value to the observed results. The methodology is equally robust and has been validated in previous studies [29]. Finally, to our knowledge, it is the first time that such a quantity of inorganic elements is measured in relation to this disease, which can contribute to broadening knowledge about a disease of such wide distribution and mortality.

## 4. Conclusions

Our study was the first to evaluate a large amount of inorganic elements in relation to stroke, including 19 inorganic elements belonging to the ATSDR’s priority pollutant list and 26 rare earth elements and other elements used in the manufacturing of high tech devices. The findings of this study indicated that patients with stroke had higher levels of lead and lower levels of bismuth, cerium, gallium and osmium. These findings provided new evidence of the potential association of dysregulated heavy metals and other elements in patients with stroke, whose ability to cross the blood brain barrier has been previously suggested. While lead was as a risk factor for stroke, barium, gold, cerium and gallium appeared as protective factors for the disease. Given the high persistence of these elements in the environment and the significant technological dependence on them, future studies are needed to elucidate the potential sources of exposure and disclose the mechanisms of action of the identified elements in the prevalence and prognosis of stroke.

## Figures and Tables

**Figure 1 toxics-08-00066-f001:**
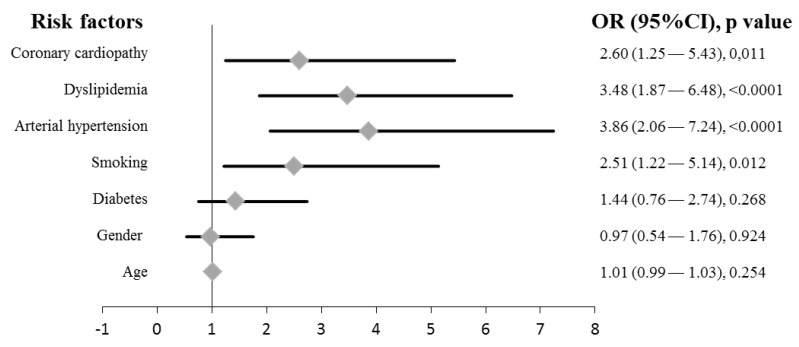
Forrest plot of odds ratios (ORs) with 95% confidence interval (CI) for factors associated with stroke. Each diamond represents the OR and the horizontal line indicates the 95% CI. For the binary logistic regression, patients were dichotomized into two groups as follows: patients who suffered stroke vs. patients who did not suffer the disease.

**Table 1 toxics-08-00066-t001:** Demographic characteristics of study participants.

	Cases *N* (%)	Controls *N* (%)	*p*-Value
All participants	92 (52.6)	83 (47.4)	
Gender			
Male	47 (51.1)	43 (51.8)	0.924 ^a^
Female	45 (48.9)	40 (48.2)	
Age (years)			
Mean ± SD	64.1 ± 12.7	61.7 ± 14.8	0.472 ^b^
Median	64	65	
Range	34–87	33–86	
Smoker (yes)	31 (33.7)	14 (16.9)	0.015 ^a^
Diabetes (yes)	25 (27.2)	29 (34.9)	0.326 ^a^
Arterial hypertension (yes)	61 (66.3)	28 (33.7)	˂0.0001 ^a^
Dyslipidemia (yes)	61 (66.3)	30 (36.1)	˂0.0001 ^a^
Coronary cardiopathy (yes)	30 (32.6)	13 (15.7)	0.013 ^a^
Barthel index			
Mean ± SD	67.2 ± 32.7	93.2 ± 16.9 ^c^	˂0.0001 ^b^
Median	75	100	
0–60 (severe dependence)	38 (41.3)	9 (11.0)	˂0.0001 ^a^
61–90 (moderate dependence)	20 (21.7)	6 (7.3)	
>90 (poor dependence/Independence)	34 (37.0)	67 (81.7)	

Abbreviations: SD, standard deviation. ^a^ Chi-squared test (two tail). ^b^ Mann–Whitney’s U test (two tail). ^c^ 1 missed data.

**Table 2 toxics-08-00066-t002:** Blood concentration (ng/mL) of inorganic elements including in the ATSDR’s priority pollutant list (2017) ^a^, among cases and controls.

	Controls (*n* = 83)			Cases (*n* = 92)			
	Frequency of Detection (%)	Median	(p5th–p95th)	Frequency of Detection (%)	Median	(p5th–p95th)	*p*-Value ^b^
Ag (silver)	79.5	0.006	(0–0.37)	80.4	0.064	(0–0.29)	0.429
As (arsenic)	100	1.69	(0.45–7.74)	100	1.61	(0.38–5.50)	0.546
Ba (barium)	100	207.0	(111.9–669.3)	100	173.5	(96.3–324.8)	˂0.0001
Be (beryllium)	8.4	0.002	(0–0.20)	12.0	0.005	(0–0.57)	0.006
Cd (cadmium)	100	0.25	(0.09–1.32)	100	0.26	(0.12–1.06)	0.359
Co (cobalt)	97.6	0.19	(0.11–0.38)	100	0.19	(0.11–0.45)	0.742
Cu (copper) ^c,d^	100	0.60	(0.42–0.89)	100	0.62	(0.49–0.80)	0.293
Hg (mercury)	98.8	3.74	(1.01–17.4)	100	3.65	(0.92–11.9)	0.793
Mn (manganese) ^c^	96.4	8.48	(0.16–14.5)	97.8	7.85	(4.55–16.8)	0.591
Ni (nickel)	92.8	1.08	(0.048–52.6)	93.5	0.98	(0.052–76.5)	0.534
Pb (lead)	100	9.03	(4.21–20.0)	100	11.2	(4.01–25.3)	0.011
Pd (palladium)	19.3	0.002	(0–0.083)	9.8	0.004	(0–0.25)	0.352
Se (selenium) ^c^	100	126.1	(73.0–205.3)	100	128.3	(79.6–169.2)	0.589
Sb (antimony)	33.7	0.027	(0.003–1.14)	26.1	0.022	(0.002–1.31)	0.151
Sr (strontium)	100	16.6	(11.2–25.0)	100	15.4	(9.93–28.7)	0.101
Th (thorium)	92.8	0.071	(0.001–0.22)	89.1	0.061	(0.001–0.12)	0.082
U (uranium)	100	0.082	(0.051–0.21)	100	0.072	(0.037–0.16)	0.024
V (vanadium)	59.0	0.011	(0.001–0.44)	35.9	0.008	(0–0.56)	0.221
Zn (zinc) ^c,d^	100	5.13	(3.7–8.06)	100	5.29	(3.77–6.53)	0.652

^a^ Complete list available at https://www.atsdr.cdc.gov/spl/. ^b^ Mann-Whitney U test (two tails). ^c^ Also considered as trace elements. ^d^ Data reported in μg/mL.

**Table 3 toxics-08-00066-t003:** Inorganic elements significantly associated with stroke.

Element	Odds Ratio	95% CI	*p*-Value ^a^
Univariate analyses			
Ba (barium)	0.34	(0.19–0.60)	<0.001
Pb (lead)	1.65	(1.09–2.50)	0.019
Au (gold)	0.81	(0.70–0.95)	0.007
Ce (cerium)	0.61	(0.42–0.90)	0.012
Ga (gallium)	0.64	(0.46–0.88)	0.007
Multivariate analyses			
Ba (barium)	0.28	(0.15–0.55)	<0.001
Pb (lead)	1.91	(1.20–3.04)	0.006
Au (gold)	0.81	(0.69–0.95)	0.011
Ce (cerium)	0.50	(0.31–0.78)	0.003
Ga (gallium)	0.58	(0.40–0.86)	0.006

^a^*p*-values were calculated by binary logistic regression. Inorganic elements are log transformed and included in the models as continuous variables. For multivariate analyses, smoking, arterial hypertension, dyslipidemia and coronary cardiopathy are included as covariables.

**Table 4 toxics-08-00066-t004:** Blood concentration (ng/mL) of rare earth elements (REE) and elements used in high tech devices ^a^, among cases and controls.

	Controls (*n* = 83)			Cases (*n* = 92)			
	Frequency of Detection (%)	Median	(p5th–p95th)	Frequency of Detection (%)	Median	(p5th–p95th)	*p*-Value ^b^
Au (gold)	57.8	0.013	(0.001–0.80)	30.4	0.007	(0–0.28)	0.001
Bi (bismuth)	86.7	0.11	(0.001–0.33)	63.0	0.085	(0–0.16)	0.001
Ce (cerium)	100	18.0	(8.02–81.7)	100	15.0	(7.23–47.2)	0.010
Dy (dysprosium)	86.7	0.017	(0–0.062)	84.8	0.018	(0–0.037)	0.459
Er (erbium)	57.8	0.002	(0–0.027)	41.3	0	(0–0.015)	0.806
Eu (europium)	45.8	0	(0–0.022)	58.7	0.007	(0–0.017)	0.047
Fe (iron) ^c,d^	100	275.5	(187.6–427.6)	100	277.8	(203.4–357.9)	0.860
Ga (gallium)	100	0.61	(0.27–4.47)	100	0.49	(0.20–1.58)	0.014
Gd (gadolinium)	69.9	0.036	(0–0.15)	63.0	0.032	(0–0.089)	0.207
Ho (holmium)	26.5	0	(0–0.010)	43.5	0	(0–0.007)	0.079
In (indium)	20.5	0	(0–0.035)	64.1	0.001	(0–0.040)	0.000
La (lanthanum)	47.0	0.010	(0.002–0.30)	22.8	0.007	(0–0.28)	0.002
Lu (lutetium)	12.0	0	(0–0.003)	7.6	0	(0–0.002)	0.425
Nb (niobium)	49.4	0.014	(0.001–0.58)	29.3	0.011	(0.001–0.57)	0.159
Nd (neodymium)	53.0	0.006	(0.001–0.28)	50.0	0.005	(0.001–0.22)	0.275
Os (osmium)	81.9	0.002	(0–0.023)	66.3	0.001	(0–0.053)	0.000
Pr (praseodymium)	48.2	0.001	(0–0.070)	50.0	0.002	(0–0.051)	0.727
Pt (platinum)	30.1	0	(0–0.014)	45.7	0	(0–0.010)	0.001
Ru (ruthenium)	60.2	0.001	(0–0.002)	22.8	0	(0–0.002)	0.000
Sm (samarium)	83.1	0.001	(0–0.067)	79.3	0.001	(0–0.045)	0.109
Sn (tin)	54.2	0.17	(0.017–4.16)	42.4	0.11	(0.018–8.58)	0.680
Ta (tantalum)	9.6	0.003	(0.001–0.28)	8.7	0.004	(0.001–0.35)	0.107
Tb (terbium)	22.9	0	(0–0.014)	8.7	0	(0–0.009)	0.410
Tm (thulium)	3.6	0	(0–0.003)	19.6	0	(0–0.003)	0.009
Y (yttrium)	54.2	0.004	(0–0.26)	56.5	0.004	(0–0.17)	0.756
Yb (ytterbium)	19.3	0	(0–0.015)	26.1	0	(0–0.012)	0.001

Abbreviations: p5th-p95th, percentiles 5 and 95 of the distribution. ^a^ Complete list available from B. Tansel et al. Environment International 98 (2017) 35-45. ^b^ Mann–Whitney U test (two tails). Significant differences are highlighted in bold. ^c^ Also considered as trace elements. ^d^ Data reported in μg/mL.

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
