# Peer review of "Association between Heavy Metals and Rare Earth Elements with Acute Ischemic Stroke: A Case-Control Study Conducted in the Canary Islands (Spain)"

_toxics, 2020, doi:10.3390/toxics8030066_

Round 1
Reviewer 1 Report
Conceptually the manuscript toxics-909598 is interested but there is still significant work that should be done. The major revision isn’t due to missing controls or needing more experimentation, but instead the presentation of the data. Potentially very interesting, points that could make this submission stand apart from others falls short. Need to increase the depth of the explanations in the discussion and conclusions.
The populations examined – there really isn’t any reason to discuss a lot about the total number of patients in the hospital, but that are not relevant to the study. Need to stick with the populations that are included
I’m not sure if people who have experienced traumatic injury and are being rehabilitated – but not stroke-related – can really be considered a ‘true’ control population. That would be someone who has not experiences stroke or trauma. I didn’t see any data that talked about exclusion criteria (smoking? Hypertension or other pre-existing condition that may contribute to the
There doesn’t seem to be any accounting for possible other external factors that could be involved in changes that were observed in inorganic matter concentration. Especially smokers v. non-smokers, which depending on the brand, could be a significant source of environmental arsenic.
I’m not really sure what the conclusions are – other than nearly 4 dozen elements were measured and examined in two groups who have suffered traumatic injury – either via stroke or some other mechanism.
There would need to be major revision and maybe a redirection of the overall hypothesis and the conclusions. For example, I am not entirely clear of the positive Pearson’s for elements like dysprosium, europium, holmium, etc., with the rare elements. If these are being reported are there any substantiating reports of the toxicity of these agents. Also, since these elements are in exceptionally low concentrations in the earth’s crust, and not used in manufacturing – how might someone be exposed? How may these be correlated to stroke – and why?
Double-check the overall writing of the submission – looking for excessively long sentences, or areas where clarity can be improved. The overall length of the paper is relatively short for the amount done, maybe increase the explanations and descriptions of WHY looking at all of these elements may be important – especially considering they are such a minor part of the exposure. For example: “The neuroprotective role of cerium has been known for years [40].” OK, but how, why? The neuroprotective role of cerium could be very interesting and to only have 1 citation – and no description of ‘how’ it is protective severely reduces the overall impact of the submission.
Author Response
Please see the attachment (Word file).

Reviewer 2 Report
General comments
This case-control study is to investigate the association of heavy metals and elements and the risk of stroke among a Spain population. The literature review is up-to-date. The methods align with the study aim. The manuscript structure are acceptable. However, the organization of result presentation is not good. Discussion is not comprehensive. Some sentences are very long, leading to difficulty understanding. There are some language problems and need linguistic expert to check the grammar and wording.
Major revision is necessary before any further consideration.
Specific comments
Abstract
Line 19-20: please clearly describe whether the OR values are obtained from univariate or multivariate analysis.
Introduction: The prevalence of stroke in US and Spain are given. It might be better to provide the average prevalence of stroke in the world, too. More introduction of inorganic elements, especially rare elements must be given.
Line 25: A clear definition of stroke should be given.
Line 27: change “is has---” to “it has---”.
Line 29-31: It would be also good to give the information of stroke incidence of Canary Islands.
Line 48-49: A clear description of inorganic elements such as the property, general health effects should be given.
Line 50-53: The statement of “ In that sense,--- contribute in some way to the stroke” is not clear. Please clarify how the presence of inorganic elements in the neurons relate to the stroke.
Patients and Methods:
Line 66-69: It is not clear whether the controls were age and gender matched with cases. Please give a clear description.
Line 69-71: for “ Patients --- entering the study.”: what about the controls? Was the same procedure as patients? Please clarify.
Line 74-75: it should give a description of how sociodemographic characteristics was collecting.
Line 117: name of Cu, Ni, Se, Fe, Ba, Zn and Sm should be given because they appear in the text first time.
Line 118-119: sentence “ The precision--- it was lower than 5% for all the elements” is not clear. What does “ it” refer to?
Line 123-124: regarding “Values below the LOQ were assigned a random value between 0 and the LOQ”, please give more detail description. How was “a random value” assigned? Normally, the values below LOW were assigned as half of LOQ value or LOW divided by square root of two.
Line 128-129: for sentence “ The correlation--- Pearson’s correlation test”, please clearly indicate which continuous variables.
line 130-131: for multivariate logistic regression analysis, which covariates were adjusted? A clear description of multivariate logistic regression including why these covariates were chosen must be given.
Results and Discussion: the results of correlation among elements are not presented and discussed!! The direction of association of element and stroke is not clearly presented. These must be considered in the manuscript revision.
Line 143-144: sentence “Although it has been published that three quarters of strokes affect patients over 65” is not clear and may lead to misunderstanding. It might be better to change “ affect” by “ occur in”
Line 144-145: sentence “ , we observed this proportion was 50.0%--- in our series.” need to be rephrased and give more explanation. What do authors mean “variables for the disease” ? And which variables are they?
Line 147: delete “In that sense”
Line 151: “ something well previously established” is not understandable and need to be rewritten.
Tabe 1: Were other sociodemographic characteristics such as race/ethnicity, family history, BMI, lifestyle (alcohol, physical activity) collected. These factors are also risk factors of stroke. In addition, the range of age should be given. At foot note, full name of BMI is given, but in Table 1, no BMI data is given. Please check!
Line 171-174: sentence “ Although serum--- the matrix of study” is very long and should be divided to two separate sentences.
Line 184-185: the description “ –that there were no---- Data not shown)” is not clear and rephrase is necessary.
Line 183-187: sentence ” Given that age--- toxic habits” is very long and need to be split. What is “ toxic habits”?? Do authors mean “ unhealthy habits”?
Line 178-190: I think it is necessary to have a table providing the correlation of demographic variables (age, smoking and gender) and inorganic elements.
Line 195-197: sentence “ In univariate analysis,--- respectively)” is not complete. What were barium and lead associated with? Please check and clarify. Results of univariate logistic regression analysis should also be given in Table 3.
Line 200: in “ Neither of these two elements was significantly influenced---“, please clearly indicate which elements are they!
Line 200-202: Where can the result of correlation between stroke factors and elements be found in the manuscript?
Line 212-216: for the statement “ In the present study--- cannot be fully understood”, authors should keep in mind that the As level measured in blood represents the As absorbed through all exposure routes. So the explanation need to be rephrased and the influence of other factors should be discussed.
Line 226-228: Was the Hg level in controls higher than case in the present study?
Line 251-252: sentence “ We did not detect significant differences with clinical variables” is not clear. Rewritten must be done.
Line 257: change “ and” to “ an”.
Line 259-261: for “ In multivariate analysis--- could play a role in stroke”, please clearly indicate gallium play what role in the stroke, protect or increase?
Line 261: add” effect” before “on stroke”.
Table 3: Univariate logistic regression analysis result should also be given in Table 3.
Conclusions: a clear conclusion of the association between elements and risk of stroke must be given!!!
Line 310: regarding RREs, give the detail element name which had lower level in cases.
Author Response
Please see the attachment (Word file).

Round 2
Reviewer 1 Report
There hare areas that have undergone extensive revision, and I will extend my gratitude to the authors for considering my previous suggestions. Still feel that there are some areas that could be improved, but nothing significant.
After going through another time, I see no reason to delay moving the paper forward.
All the best
Reviewer 2 Report
General comments
The authors considered the comments of reviewer and made the corresponding revision. The manuscript improve very much. There is only a few comments. After further revision, the manuscript is suitable to publish.
Specific comments
Concerning abbreviation of 95% confidential interval, typically use “95% CI”, not “ CI 95%”, please consider to change in the text.
Line 92: a sentence “ The control group consisted of patients admitted to the rehabilitation service for other causes” should be added before “ The final number of contril subjects was 83.”
line 164-165: A brief description of how these covariates were chosen must be given.
Line 182-184: sentence “ Due to ---- and stroke” belong to Statistical analysis and should consider to move.
Line 247-248: when reading “ Thus, median values--- 1.18ng/mL, respectively”, the reader will think these values are from the same population of the present study. But they are result of a Chinese population. Rewritten is necessary.
